# ACPO: Adaptive Curriculum Policy Optimization for Aligning Vision-Language Models in Complex Reasoning

## Abstract

Aligning large-scale vision-language models (VLMs) for complex reasoning via reinforcement learning is often hampered by the limitations of existing policy optimization algorithms, such as static training schedules and the rigid, uniform clipping mechanism in Proximal Policy Optimization (PPO). In this work, we introduce Adaptive Curriculum Policy Optimization (ACPO), a novel framework that addresses these challenges through a dual-component adaptive learning strategy. First, ACPO employs a dynamic curriculum that orchestrates a principled transition from a stable, near on-policy exploration phase to an efficient, off-policy exploitation phase by progressively increasing sample reuse. Second, we propose an Advantage-Aware Adaptive Clipping (AAAC) mechanism that replaces the fixed clipping hyperparameter with dynamic, sample-wise bounds modulated by the normalized advantage of each token. This allows for more granular and robust policy updates, enabling larger gradients for high-potential samples while safeguarding against destructive ones. We conduct extensive experiments on a suite of challenging multimodal reasoning benchmarks, including MathVista, LogicVista, and MMMU-Pro. Results demonstrate that ACPO consistently outperforms strong baselines such as DAPO and PAPO, achieving state-of-the-art performance, accelerated convergence, and superior training stability.

## 1 Introduction

Large language models (LLMs) such as LLaMA Touvron et al. (2023) and GPT-4 OpenAI (2023) have revolutionized natural language processing, exhibiting strong few-shot generalization and reasoning capabilities. Extending this paradigm, vision-language models (VLMs) integrate visual perception with language understanding, enabling tasks such as captioning, visual question answering (VQA), and multimodal reasoning. Representative examples include CLIP Radford et al. (2021), Flamingo Alayrac et al. (2022), Kosmos-1 Huang et al. (2023), Gemini Author (2023), and the Qwen-VL family Team (2025). These models demonstrate that multimodal LLMs can serve as general-purpose agents capable of tackling complex reasoning tasks across domains.

Despite their strong pretraining capabilities, LLMs and VLMs typically require an alignment stage before deployment Kirk et al. (2023), to ensure outputs are faithful, safe, and aligned with human intent. Reinforcement learning from human feedback (RLHF) has emerged as the predominant framework for this alignment Ouyang et al. (2022); Kaufmann et al. (2023). Classical policy optimization methods, such as Proximal Policy Optimization (PPO) Schulman et al. (2017), help stabilize training, but their static schedules and uniform clipping mechanisms are often suboptimal for token-level updates in large models. This has motivated several refinements: Direct Preference Optimization (DPO) Rafailov et al. (2023) simplifies reward modeling by integrating preference signals directly into policy gradients; Group Relative Policy Optimization (GRPO) Shao et al. (2024); Guo (2025) leverages group-wise comparisons to enhance sample efficiency; and multimodal extensions like DAPO Yu et al. (2025) and PAPO Huang et al. (2024) adapt these methods to vision-language reasoning. While effective, these approaches still rely on rigid clipping and static hyperparameters, which can limit learning efficiency and introduce instability when encountering high-variance or noisy rewards.

To address these limitations, we introduce **Adaptive Curriculum Policy Optimization (ACPO)**, a novel RL framework that adapts its learning strategy dynamically to the evolving capabilities of the model. ACPO employs a dual-component adaptive learning strategy designed to improve both stability and sample efficiency.

First, a **dynamic curriculum policy** orchestrates a principled transition between learning phases. ACPO begins with a stable, near-on-policy exploration phase, using frequent data refreshes and short reuse windows to ensure robust gradient estimation. As training progresses, the curriculum transitions to an off-policy exploitation phase, gradually increasing sample reuse to allow intensive fine-tuning on high-quality data without risking overfitting or catastrophic forgetting.

Second, we introduce an **Advantage-Aware Adaptive Clipping (AAAC)** mechanism, which refines PPO's update rule by replacing the fixed clipping threshold with dynamic, sample-wise bounds modulated by the normalized advantage of each token. High-advantage samples are allowed wider updates, while low- or negative-advantage samples are conservatively constrained, improving gradient allocation and policy robustness.

We evaluate ACPO on several challenging multimodal reasoning benchmarks, including Math-Vista Luo et al. (2023), LogicVista Wang et al. (2024), DynaMath Zhang et al. (2024b), and MMMU-Pro Yu et al. (2023). Experimental results show that ACPO consistently outperforms strong baselines such as DAPO and PAPO, achieving faster convergence, improved training stability, and state-of-the-art performance across all tasks.

Our contributions can be summarized as follows:

- We propose a dynamic curriculum framework that balances on-policy exploration with off-policy exploitation, allowing training to adapt as the model's capabilities evolve.
- We introduce AAAC, which replaces PPO's fixed clipping with advantage-aware, sample-wise bounds for more granular and robust policy updates.
- Extensive experiments demonstrate that ACPO achieves state-of-the-art performance and accelerated convergence on multiple complex multimodal reasoning benchmarks.

## 2 RELATED WORK

### 2.1 REINFORCEMENT LEARNING FROM HUMAN FEEDBACK

RLHF has become the dominant paradigm for aligning large language models with human preferences Ouyang et al. (2022); Kaufmann et al. (2023). Early approaches typically employ Proximal Policy Optimization (PPO) Schulman et al. (2017) for training stabilization, with a static clipping mechanism adopted to constrain policy updates. While effective, PPO's uniform clipping can lead to suboptimal updates, suppressing high-advantage signals or failing to constrain harmful updates, which may cause instability or entropy collapse.

Building upon this, subsequent algorithms have sought to refine the optimization process. GRPO introduced a group-based reward formulation that aggregates responses per prompt and computes a shared advantage signal across all generated outputs, which improves training stability by reducing variance in reward estimation. It also employs a token-level Kullback-Leibler (KL) penalty to prevent excessive deviation from the reference policy at the sequence level, thereby mitigating mode collapse while preserving fine-grained control over generation. DAPO introduced several key improvements to enhance stability and sample efficiency. To counter entropy collapse, DAPO proposed the clip-higher strategy, which asymmetrically increases the upper clipping bound to encourage exploration. To address vanishing gradients for prompts with near-perfect or zero accuracy, it introduced a Dynamic Sampling mechanism to filter out these less informative instances. DAPO also incorporated token-level loss and a soft penalty for overlong responses to further stabilize training.

Furthermore, Macro-Action RLHF (MA-RLHF) Chai (2024) introduces macro actions, such as token sequences or higher-level language structures, to reduce credit assignment issues over long horizons and improve learning efficiency. Contrastive reward mechanisms Shen (2024) reduce uncertainty in reward models and encourage improvement beyond baseline performance, mitigating variance issues in PPO. Personalized RLHF approaches Poddar (2024) capture diverse user preferences using variational methods, enabling personalized reward modeling and better performance

across different user populations. Unsupervised RLHF Solway (2024) leverages signals derived automatically from data to provide negative guidance, enabling fine-grained model adjustment without the need for additional human feedback. Reward ensemble methods Zhang (2024) combine multiple reward models to enhance prediction accuracy, addressing errors caused by limited training data in conventional RLHF.

While these efforts have broadened the horizons of reward modeling, policy updates, and personalized alignment, achieving robust training stability and high sample efficiency remains a central challenge. To this end, we introduce ACPO, a method that incorporates a dynamic curriculum and Advantage-Aware Adaptive Clipping to directly tackle these issues.

## 2.2 CURRICULUM LEARNING FOR REINFORCEMENT LEARNING

Curriculum learning (CL) enhances training by organizing tasks in a progressive sequence, starting with simpler ones and gradually increasing complexity. This approach aims to improve both learning efficiency and generalization, Bengio et al. (2009). In RL, CL techniques include methods like task sorting by difficulty Wang et al. (2019); Justesen et al. (2018), teacher-student models that adaptively select tasks based on the learner's progress Portelas et al. (2020), and self-play strategies that create curricula through agent competition Sukhbaatar et al. (2017).

Although CL has been extensively explored in traditional RL, its use in RLHF for LLMs remains limited. Current approaches typically rely on staged training with predefined difficulty levels Wen et al. (2025); Luo et al. (2025); Song et al. (2025) or online filtering techniques that sample and discard data until that rewards fall within a specific range Bae et al. (2025); Yu et al. (2025). However, these methods often lack adaptability due to dynamic difficulty levels in each batch of the training data.

In contrast, our framework actively guides the learning trajectory by fully considering the evolving nature of the training process. It employs a dual-component mechanism: a dynamic frequency control scheduler that orchestrates the transition from stable on-policy updates to efficient off-policy sample reuse, and a course-aware sample screening process that progressively increases the difficulty of training data. This structured approach ensures the model first masters foundational knowledge before focusing on more challenging examples, leading to more robust and efficient convergence.

## 3 METHOD

### 3.1 OVERVIEW

Aligning VLMs for complex reasoning via RLHF has become a predominant paradigm. While recent algorithms like GRPO and DAPO have achieved significant gains in sample efficiency and performance, they are often limited by static training schedules and a fixed clipping threshold in PPO. This rigid, one-size-fits-all mechanism can be suboptimal, either suppressing beneficial policy updates or failing to prevent destructive ones, which leads to training instability and limits the model's potential. To overcome these challenges, we introduce ACPO, a novel framework illustrated in Fig 1. Our approach features two key innovations: a dynamic curriculum that intelligently transitions training from a stable on-policy to an efficient off-policy regime (Fig 1(B)), and a novel sample-wise adaptive clipping mechanism that modulates optimization bounds on a per-sample basis according to its advantage (Fig 1(C)). This dual approach significantly enhances training stability and convergence efficiency, leading to state-of-the-art performance.

#### 3.1.1 STRATEGIC GATING SAMPLING

To enhance training stability and focus the model on high-quality signals, ACPO first employs a strategic sample gating mechanism. At each training step $t$, for a candidate batch of queries $\mathcal{B}_t = \{q_j\}_{j=1}^{M}$, we generate responses using the reference policy $\pi_{\theta_{\text{old}}}$. This batch is then filtered to produce a high-quality subset, $\mathcal{B}_{\text{valid}}$, based on reward and diversity criteria:

$$\mathcal{B}_{\text{valid}} = \left\{ q \in \mathcal{B}_t \mid 0 < \sum_{i=1}^{G} \mathbb{I}(R(o_i) > \tau) \leq N_{\max} \right\} \tag{1}$$

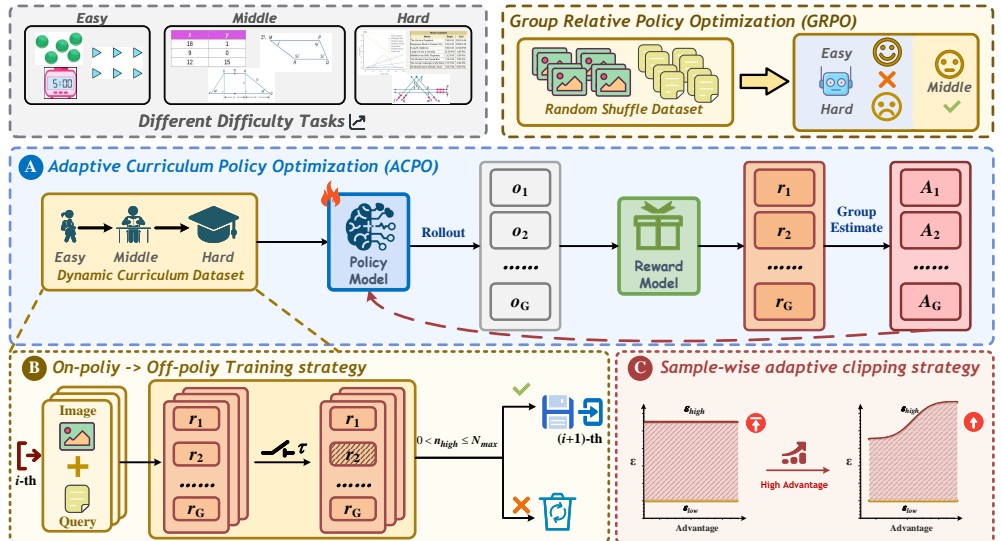

Figure 1: Overview of ACPO. Unlike GRPO, ACPO removes the KL divergence constraint. Module B introduces dynamic curriculum sampling, where the $i-th$ iteration selects moderately difficult samples based on threshold $\tau$ and $N_{max}$, which then proceed to $(i+1)-th$ iteration . Module C adds advantage-based clipping, enabling safer, more effective updates for high-advantage samples.

where $\tau$ is the minimum reward threshold and $N_{\max}$ is the maximum number of high-reward responses per query, which encourages diversity. Only queries that elicit a sufficient number of high-reward responses are retained in $\mathcal{B}_{\text{valid}}$ for the subsequent optimization phase.

### 3.1.2 ON-POLICY TO OFF-POLICY PHASE TRANSITION

After identifying the valid samples, ACPO uses its dynamic curriculum to manage the stability-efficiency trade-off, governed by the adaptive reuse count $K(t)$.

The GRPO objective's expectation is taken over the original, unfiltered batch $\mathcal{B}_t$. The crucial link to the gating mechanism is established by incorporating an indicator function, $\mathbb{I}(q \in \mathcal{B}_{\text{valid}})$, which effectively masks out the loss for any sample that did not meet the gating criteria:

$$
J_{\text{GRPO}}(\theta) = \mathbb{E}_{\mathbf{q} \sim \mathcal{B}_t, \{o_i\} \sim \pi_{\theta_{\text{old}}}(\cdot|q)} \left[ \frac{1}{G} \sum_{i=1}^{G} \frac{1}{|o_i|} \sum_{t=1}^{|o_i|} \right.
$$
$$
\left. \left( \min \left( r_{i,t}(\theta)\hat{A}_{i,t}, \ \text{clip}\left(r_{i,t}(\theta), 1-\epsilon, 1+\epsilon\right)\hat{A}_{i,t} \right) - \beta D_{\text{KL}}\left(\pi_\theta \,\|\, \pi_{\text{ref}}\right) \right) \right] \tag{2}
$$

where $r_{i,t}(\theta)$ is the probability ratio $\frac{\pi_\theta(o_{i,t}|q,o_{i,<t})}{\pi_{\theta_{\text{old}}}(o_{i,t}|q,o_{i,<t})}$ and $\hat{A}_{i,t}$ is the advantage estimate. Crucially, the expectation is now taken over queries $q$ drawn from the gated batch $\mathcal{B}_{\text{valid}}$ (from Eq. 1), ensuring that only high-quality samples contribute to the gradient.

Instead of performing a fixed number of updates, ACPO performs $K(t)$ optimization steps using the objective in Eq. 2, where $K(t)$ adapts with training progress:

$$
K(t) = \max\left(1, \left\lceil \frac{N \cdot t}{T} \right\rceil\right) \tag{3}
$$

where $N$ is the maximum reuse count, $t$ is the current training step, and $T$ is the total duration. This curriculum creates a principled transition through three distinct phases:

- **On-policy Exploration Phase** ($t \ll T$)**:** When $K(t) \approx 1$, the model prioritizes stable learning on fresh, high-quality data to build a robust policy foundation.

- **Balanced Transition Phase:** As $K(t)$ grows linearly, the strategy gradually anneals towards off-policy learning, increasing sample reuse as the policy stabilizes.
- **Off-policy Exploitation Phase ($t \to T$):** When $K(t) \to N$, the model intensively fine-tunes its policy on the most valuable gated samples, maximizing data utility to accelerate final convergence.

## 3.2 ADVANTAGE-AWARE ADAPTIVE CLIPPING

A primary limitation of the standard PPO algorithm is its reliance on a fixed clipping hyperparameter, $\epsilon$, which applies a uniform update constraint to all samples regardless of their learning potential. This can either stifle progress on high-quality samples or fail to prevent destructive updates from noisy ones.

To overcome this, ACPO introduces an **Advantage-Aware Adaptive Clipping** mechanism. Instead of a static bound, the upper clipping range is dynamically modulated by the magnitude of the sample's advantage, allowing for a more granular and intelligent policy update. The ACPO objective is formulated as:

$$
J_{\text{ACPO}}(\theta) = \mathbb{E}_{q \sim \mathcal{B}_{\text{valid}},\, \{o_i\} \sim \pi_{\theta_{\text{old}}}(\cdot|q)} \left[ \frac{1}{\sum_{i=1}^{G} |o_i|} \sum_{i=1}^{G} \sum_{t=1}^{|o_i|} \right.
$$

$$
\left. \min\left( r_{i,t}(\theta)\hat{A}_{i,t},\; \text{clip}\left( r_{i,t}(\theta), 1 - \epsilon_{\text{low}}, 1 + \epsilon_{\text{high}}(\hat{A}_{i,t}) \right) \hat{A}_{i,t} \right) \right]
$$

(4)

The key innovation lies in the upper clipping bound, $\epsilon_{\text{high}}$, which is no longer a fixed value but a function of the token-level advantage $\hat{A}_{i,t}$:

$$
\epsilon_{\text{high}}(\hat{A}_{i,t}) = \epsilon_{\text{high}}^0 + \delta \cdot \tilde{A}_{i,t}
$$

(5)

where $\epsilon_{\text{high}}^0$ is a baseline clipping value and $\delta$ is a scaling factor controlling the sensitivity to the advantage. The term $\tilde{A}_{i,t}$ represents the normalized advantage, which is transformed from an un-

---

**Algorithm 1** Dynamic Curriculum Policy Optimization (ACPO)

---

1: **Input**: Initial policy $\pi_{\theta_{\text{init}}}$, reward model $r_\phi$, prompt dataset $\mathcal{D}$.
2: **Hyperparameters**: Max reuse count $N$, outer iterations $I$, training steps per iteration $T$, batch size $M$, clipping baseline $\epsilon_{\text{high}}^0$, sensitivity $\delta$, reward threshold $\tau$, diversity count $N_{\text{max}}$.
3: **Output**: Optimized policy $\pi_\theta^*$.
4: Initialize policy $\pi_\theta \leftarrow \pi_{\theta_{\text{init}}}$.
5: **for** iteration $i = 1$ to $I$ **do**
6:     Set reference policy for KL penalty $\pi_{\text{ref}} \leftarrow \pi_\theta$.
7:     **for** training step $t = 1$ to $T$ **do**
8:         Sample a batch of prompts $\mathcal{B}_t = \{q_j\}_{j=1}^{M} \sim \mathcal{D}$.
9:         Set old policy for sampling $\pi_{\theta_{\text{old}}} \leftarrow \pi_\theta$.
10:        Generate responses $\{o_i\}_{i=1}^{G} \sim \pi_{\theta_{\text{old}}}(\cdot|q)$ for each $q \in \mathcal{B}_t$.
11:        Construct the valid batch $\mathcal{B}_{\text{valid}} \subseteq \mathcal{B}_t$ using the gating criteria in Eq. 1.
12:        Compute rewards $R(o_i)$ for all responses using $r_\phi$.
13:        Compute advantages $\hat{A}_{i,t}$ for each token in all responses.
14:        {Begin adaptive update phase}
15:        Determine curriculum reuse count $K(t) \leftarrow \max(1, \lceil N \cdot t/T \rceil)$ using Eq. 3.
16:        **for** update epoch $k = 1$ to $K(t)$ **do**
17:           Compute loss $L(\theta)$ on batch $\mathcal{B}_t$ using the full ACPO objective $J_{\text{ACPO}}$ from Eq. 2.
18:           {The objective implicitly masks invalid samples and uses adaptive clipping.}
19:           Update policy parameters $\theta \leftarrow \text{optimizer\_step}(\theta, \nabla_\theta L(\theta))$.
20:        **end for**
21:     **end for**
22: **end for**
23: **return** optimized policy $\pi_\theta^* \leftarrow \pi_\theta$.

---

bounded range to $[0, 1]$ using the error function (erf):

$$\tilde{A}_{i,t} = \frac{1}{2}\left(1 + \text{erf}\left(\frac{\hat{A}_{i,t}}{\sqrt{2}\sigma_A}\right)\right) \tag{6}$$

where $\sigma_A$ is the standard deviation of the advantages in the batch, used for scaling. This formulation establishes a fine-grained, sample-wise control over the optimization landscape. High-advantage samples are rewarded with a significantly wider clipping range, enabling larger and more confident policy updates that capitalize on strong learning signals. Conversely, low- or negative-advantage samples are met with a conservative bound that shields the policy from noisy or potentially destructive gradients. In essence, this mechanism allows ACPO to dynamically allocate its gradient budget—accelerating convergence by exploiting high-potential updates while preserving the stability crucial for complex reasoning tasks.

The entire process, which integrates strategic data gating with an adaptive update curriculum, is summarized in Alg. 1. The pseudocode outlines the complete training loop, from data sampling and filtering to the dynamically scheduled, advantage-aware policy updates.

## 4 EXPERIMENTS

### 4.1 EXPERIMENTAL SETUP

All experiments were conducted on four servers, each equipped with eight H20 GPUs. The training process utilized the DeepSpeed Zero2 (Rasley et al., 2020) configuration to optimize memory usage and efficiency. We based our models on the Qwen2.5-VL-3B (Bai et al., 2025), training them on the ViRL39K dataset Wang et al. (2025) with a learning rate of 1e-6 using direct reinforcement learning. Our proposed ACPO method was compared against standard baselines, including DAPO and PAPO. Since ACPO blends on-policy and off-policy approaches, we included both on-policy and off-policy DAPO baselines in our evaluation for a comprehensive comparison.

#### 4.1.1 EVALUATION

To comprehensively evaluate the effectiveness of our method, we conducted experiments on seven public benchmarks covering diverse reasoning domains. These include: Geometry3K Lu et al. (2021), MathVerse , MathVerse-V Zhang et al. (2024a), and We-Math Qiao et al. (2024) for mathematical and geometric reasoning; MMMU-Pro Yue et al. (2024) for multi-discipline multimodal reasoning; LogicVista Xiao et al. (2024) for logical reasoning; and Counting Li et al. (2023) for counting tasks. Evaluation was based on exact match between model predictions and ground-truth answers. We report the average accuracy@8 across all benchmarks with a reasoning temperature of 1.0. Datasets requiring free-form responses or those evaluated by LLM-based judges were excluded from this study.

#### 4.1.2 MAIN RESULTS

The superior performance of ACPO, as evidenced in Tab. 1 and 2, can be directly attributed to its two core methodological innovations: the dynamic on-policy to off-policy curriculum and the advantage-aware adaptive clipping mechanism. The consistent gains across both 3B and 7B scales—particularly in general reasoning tasks like MathVerse, Geo3k, and We-Math—reflect the effectiveness of ACPO's strategic sample gating and phased training schedule. By initially operating in a stable on-policy regime, ACPO avoids the early-stage instability that often plagues off-policy methods, allowing the policy to establish a reliable foundation. As training progresses, the linear increase in reuse count $K(t)$ enables efficient exploitation of high-reward, gated samples, which explains the pronounced improvements in tasks requiring compositional or abstract reasoning where high-quality supervision signals are sparse but critical. This curriculum-aware reuse not only enhances data efficiency but also ensures that the model refines its behavior on the most informative examples during the final exploitation phase, directly contributing to ACPO's leading average scores in both reasoning categories.

Furthermore, the advantage-aware adaptive clipping mechanism provides a fine-grained control over policy updates that standard PPO's fixed $\epsilon$ cannot match. In complex multimodal settings, where

Table 1: Comparative performance evaluation across vision-dependent and general tasks in 3B model sizes. **RED BOLD** indicates the best performance, and UNDERLINED indicates the second-best performance.

| Model | Overall | Vision-Dependent Multimodal Reasoning | | | | | General Multimodal Reasoning | | | |
|---|---|---|---|---|---|---|---|---|---|---|
| | Average(@8) | MathVerse-V | MMMU-Pro | Counting | LogicVista | AVG | MathVerse | Geo3k | We-Math | AVG |
| DAPO-Off$_{3B}$ | 44.51% | 46.37% | 26.76% | 72.06% | 37.36% | 45.63% | 49.72% | 24.60% | 54.70% | 43.01% |
| DAPO-On$_{3B}$ | 47.68% | 48.32% | 28.69% | **73.88%** | 39.07% | 47.49% | 51.96% | 32.57% | 59.30% | 47.94% |
| PAPO$_{3B}$ | 47.26% | 49.04% | 29.31% | 65.88% | **41.41%** | 46.41% | 54.79% | 31.82% | 58.57% | 48.39% |
| ACPO$_{3B}$ | **49.90%** | **53.63%** | **29.60%** | 72.75% | 41.14% | **49.28%** | **57.41%** | **33.13%** | **61.67%** | **50.74%** |

Table 2: Comparative performance evaluation across vision-dependent and general tasks in 7B model sizes. **RED BOLD** indicates the best performance, and UNDERLINED indicates the second-best performance.

| Model | Overall | Vision-Dependent Multimodal Reasoning | | | | | General Multimodal Reasoning | | | |
|---|---|---|---|---|---|---|---|---|---|---|
| | Average(@8) | MathVerse-V | MMMU-Pro | Counting | LogicVista | AVG | MathVerse | Geo3k | We-Math | AVG |
| DAPO-Off$_{7B}$ | 50.82% | 51.51% | 30.25% | 89.25% | 38.17% | 52.30% | 56.00% | 22.59% | 53.94% | 44.18% |
| DAPO-On$_{7B}$ | 56.05% | 57.51% | 35.20% | 87.19% | 44.24% | 56.04% | 61.62% | 32.51% | 63.36% | 52.50% |
| PAPO$_{7B}$ | 59.15% | 64.97% | 36.63% | **89.81%** | 46.07% | **59.37%** | **69.53%** | 40.25% | 66.79% | 58.85% |
| ACPO$_{7B}$ | **60.07%** | **65.10%** | **37.10%** | 82.12% | **47.93%** | 58.06% | 68.65% | **41.58%** | **69.15%** | **59.79%** |

token-level advantages vary significantly—e.g., a correct geometric deduction in Geo3k may yield high advantage, while a misaligned visual reference in Counting may produce low or negative advantage—ACPO dynamically widens the clipping bound for high-advantage tokens, enabling aggressive updates where the signal is strong, while constraining updates for ambiguous or noisy samples. This explains why ACPO achieves the best results on high-stakes benchmarks like We-Math (61.67% at 3B, 69.15% at 7B) and Geo3k (33.13% at 3B, 41.58% at 7B), where precise, confident reasoning steps are essential. The adaptive clipping thus acts as an implicit "reasoning confidence modulator," aligning optimization intensity with the reliability of each learning signal—ultimately yielding a more robust, scalable, and consistently top-performing policy across diverse multimodal reasoning challenges.

Fig.2 (a) and (c) show the cumulative reward curves of ACPO$_{3B}$ and the baseline DAPO$_{3B}$ under off-policy and on-policy settings, respectively. Both methods exhibit rapid initial performance improvement, indicating strong learning capability in the early training stages. However, under the off-policy setting, DAPO demonstrates significant reward fluctuations and achieves a notably lower convergence value compared to ACPO, suggesting an unstable policy update process that hinders long-term performance growth. In the on-policy setting, although DAPO reaches a convergence level close to that of ACPO, its reward trajectory remains highly volatile, indicating a lack of robustness in the optimization process. In contrast, ACPO consistently exhibits smoother convergence and higher final performance across both settings, highlighting its superior stability and generalization capability under different data collection strategies.

Fig.2 (b) and (d) present the corresponding clip ratio dynamics of both methods (for the 3B models). It can be observed that under the off-policy setting, the clip ratio of DAPO progressively increases during training, implying that a large portion of advantage signals are clipped. This restricts the magnitude of policy updates and prevents the model from fully leveraging the guidance of high-reward action directions. In contrast, ACPO maintains a consistently low clip ratio, indicating that it preserves more genuine advantage signals and allows more aggressive updates along high-advantage trajectories. This enables ACPO to explore higher-performance regions in the policy space, thereby achieving a superior performance upper bound. This behavior aligns with the superior convergence observed in Fig.2 (a), further demonstrating the effectiveness of ACPO's policy update mechanism.

### 4.1.3 ABLATION STUDY ANALYSIS

As shown in Table 1 and 3, we conduct an ablation study to evaluate the effectiveness of the AAAC mechanism. The results demonstrate that removing AAAC (i.e., ACPO w/o AAAC) leads to a performance degradation across multiple benchmark tasks, particularly in vision-dependent and general multimodal reasoning scenarios. Specifically, the overall accuracy drops from 49.90%

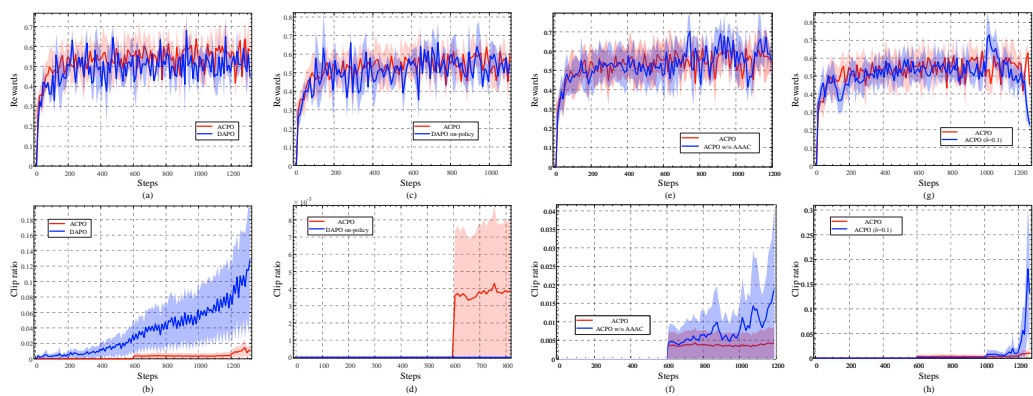

Figure 2: Training Dynamics of Reward and Clip Ratio in Ablation and Baseline RL Experiments.

Table 3: Ablation study of AAAC and scaling factor in 3B model size. **RED BOLD** indicates the best performance, and UNDERLINED indicates the second-best performance.

| Model | Overall | Vision-Dependent Multimodal Reasoning | | | | | General Multimodal Reasoning | | | |
|---|---|---|---|---|---|---|---|---|---|---|
| | Average(@8) | MathVerse-V | MMMU-Pro | Counting | LogicVista | AVG | MathVerse | Geo3k | We-Math | AVG |
| ACPO w/o AAAC | 48.74% | 53.27% | 28.91% | 68.88% | **41.39%** | 48.11% | 56.36% | 31.01% | 61.39% | 49.59% |
| ACPO$_{\delta=0.10}$ | 47.28% | 52.38% | 26.95% | 58.88% | 40.72% | 44.73% | 56.00% | **33.78%** | **62.28%** | 50.69% |
| ACPO$_{\delta=0.03}$ | 47.52% | 50.99% | 27.60% | 68.19% | 40.97% | 46.94% | 54.24% | 29.01% | 61.66% | 48.30% |
| ACPO$_{\delta=0.05}$ | **49.90%** | **53.63%** | **29.60%** | **72.75%** | 41.14% | **49.28%** | **57.41%** | 33.13% | 61.67% | **50.74%** |

(ACPO) to 48.74% (ACPO w/o AAAC), confirming that AAAC plays a crucial role in enhancing the model's reasoning capability.

Fig.2 (e) and (f) present the ablation results after removing the AAAC module. It can be observed that after switching to the off-policy training setting, the model without AAAC exhibits a significant increase in the clipping ratio of advantage signals, accompanied by intensified fluctuations in the reward curve. This outcome is attributed to the curriculum learning mechanism, which continuously introduces more difficult samples during training. Without the AAAC module to effectively learn from such samples, the model fails to improve its performance when exposed to a large number of challenging instances; instead, it experiences a degradation in capability. These results fully demonstrate the importance of the AAAC mechanism in handling difficult samples, maintaining training stability, and enhancing overall performance.

In the ACPO algorithm, the clipping range of AAAC is set to 0.05. Fig.2 (g) and (h) present the experimental results when the AAAC range is expanded to 0.1. As shown in Fig.2 (g), during the early training phase (up to approximately 1000 steps), the reward curves under both settings are similar, indicating comparable learning behavior initially. However, beyond 1000 steps, the model with the larger AAAC range of 0.1 exhibits a noticeable decline in reward, demonstrating clear performance degradation. This behavior can be attributed to the overestimation of high-advantage signals caused by the excessively wide AAAC range, which results in policy updates that deviate too drastically from the reference policy. Such excessive deviation prevents the model from effectively learning useful policy information, ultimately leading to unstable or even divergent training. Further insights can be drawn from the clip ratio dynamics in Fig.2 (h). Although a larger AAAC range should theoretically allow more aggressive updates, a higher clip ratio is observed in practice. This indicates that when encountering difficult samples, the model fails to capture meaningful environmental feedback, still generating high advantage estimates that trigger more frequent clipping. This reflects instability in the policy optimization process. In conclusion, the configuration of the AAAC range significantly affects both training stability and learning efficiency. While intended to promote exploration, an excessively large range may lead to policy divergence and learning failure due to overly aggressive updates.

Furthermore, the ablation results in Tab. 3 further validate the significant impact of the AAAC clipping range on the model's final performance. When $\delta = 0.10$, the overly aggressive update strategy fails to improve performance and instead leads to training instability, hindering convergence. In contrast, when $\delta = 0.03$, the update rule becomes excessively conservative, limiting the model's exploratory capacity and impeding effective learning. Through extensive empirical evaluation, we find that $\delta = 0.05$ strikes an optimal balance between update magnitude and training stability, effectively trading off exploration and exploitation, and thereby achieving the best overall performance.

## 5 CONCLUSION

In this work, we present ACPO, a novel framework designed to overcome the limitations of static training schedules and fixed optimization boundaries inherent in prior reinforcement learning methods. The core innovation of ACPO lies in its dual adaptive mechanisms: a dynamic curriculum that orchestrates a smooth transition from stable exploration to efficient exploitation by intelligently scheduling data, and our proposed AAAC, which replaces the fixed clipping threshold with sample-wise dynamic bounds to enable more granular and effective policy updates. Extensive experiments validate the superiority of our approach: ACPO not only achieves a state-of-the-art average accuracy of 49.90% across multiple complex multimodal reasoning benchmarks, significantly outperforming strong baselines like DAPO and PAPO, but also exhibits faster convergence and exceptional training stability. These advantages demonstrate that ACPO establishes a more efficient, robust, and adaptive optimization paradigm for the alignment of large-scale vision-language models.

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
