# A APPENDIX

As showen in Tab. 4, in the initial training phase, the model is exposed primarily to "Easy" problems. These tasks are straightforward and involve direct application of basic concepts. For example, determining whether Addison can afford both items requires only simple arithmetic and comparison. Calculating the area ratio of similar triangles relies solely on a fundamental geometric property.

Through such elementary exercises, the model quickly learns core mathematical operations and basic geometric principles. This process establishes an essential knowledge foundation. It mirrors how students first learn arithmetic and simple geometry to build initial cognitive frameworks.

As training advances, the model encounters "Middle" difficulty problems. These require integrated application of knowledge and preliminary logical reasoning. For instance, finding a circle's radius given a perpendicular chord demands combining the perpendicular chord theorem with the Pythagorean theorem.

Another example is calculating the surface area of a composite solid. This requires careful spatial visualization and systematic counting of exposed faces. Algorithmic problems at this level involve tracing execution steps to deduce outputs.

Table 4: Samples of varying difficulty across curriculum learning stages.

| Type | Image | Query |
|---|---|---|
| Eazy |  | With \$1,762.00, can Addison afford to purchase both a pair of designer shoes and a designer scarf? |
| Eazy |  | $\triangle ABC$ and $\triangle DEF$ are similar with point O as the center of similarity, and $\frac{OD}{OA} = \frac{1}{2}$. Then $\frac{S_{\triangle DEF}}{S_{\triangle ABC}} =$? |
| Eazy |  | What was Google's ranking out of 100 ACSI index points in 2020? |
| Middle |  | $AB$ is the diameter, $CD$ is a chord, and $CD \perp AB$, $CD = 4\sqrt{2}$ and $AE = 2$, the radius is ___. |
| Middle |  | A geometric solid is formed by 5 cubes with an edge length of 1. The surface area of this geometric solid is ___. |
| Middle |  | In the following algorithm, the output value of $i$ is ___. |
| Difficult |  | The maximum number of small cubes that make up this geometric body is ___. |
| Difficult |  | $AB = 6$ is the diameter. The semicircle is rotated $45°$ clockwise around point $A$. What is the area of the shaded region? |
| Difficult |  | According to the following algorithm statements, when the input $x$ is 60, the output value of $y$ is? |

At this stage, the model must synthesize previously acquired knowledge. It develops logical structuring skills and enhances comprehensive problem-solving abilities. This phase resembles students progressing to multi-concept exercises.

In the final training stage, the model faces "Difficult" problems. These feature high complexity and substantial cognitive challenges. For example, determining the maximum number of unit cubes in a solid requires advanced spatial reasoning and exploring various combinatorial possibilities.

Another challenging task involves calculating shaded areas after rotational transformations. This demands understanding geometric transformations and applying area decomposition strategies. Algorithmic problems may involve nested conditionals or iterative computations.

At this level, the model must deeply integrate all previously learned knowledge and skills. It demonstrates advanced capabilities like rigorous spatial imagination and multi-step logical deduction. This stage is analogous to students tackling contest-level problems requiring innovative application of concepts.