# OpenReview forum: "ACPO: Adaptive Curriculum Policy Optimization for Aligning Vision-Language Models in Complex Reasoning"
_ICLR.cc/2026/Conference — ICLR 2026 Conference Desk Rejected Submission_

### Official Review · Reviewer_3pG4 · 2025-10-27

**Soundness:** 2
**Presentation:** 2
**Contribution:** 2
**Rating:** 4
**Confidence:** 4

**Summary:**

This paper proposes ACPO, a novel reinforcement learning method for VLM alignment in reasoning tasks.
Specifically, ACPO leverages the novel dynamic curriculum for a trade-off the on-policy exploration and off-policy exploitation. Additionally, ACPO introduces a novel advantage-aware clipping mechanism for robust training.
Experiments on different benchmarks demonstrate that ACPO outperforms existing strong baselines, including DAPO.

**Strengths:**

This paper introduces ACPO with interesting modifications and conducts experiments in different challenging benchmarks.

**Weaknesses:**

1. There are no illustrations of module A in Fig. 1.
2. It is improper and unclear to use the term "dynamic curriculum", which misleads the reviewer into knowing the exact illustration of the proposed method in section 3.1.2. Eq. (3) is basically an early-stop criterion in on-policy RL methods (e.g., PPO sometimes use entropy as the early-stop criterion).
3. The underlying motivation of AAAC is similar to existing methods, like clip higher. It would be great to compare AAAC and clip higher using the same baseline (e.g., DAPO).
4. There are no ablation studies to demonstrate the efficiency of the dynamic curriculum. For instance, the performance comparison with fixed optimisation steps. Or can the dynamic curriculum make ACPO have comparable performance while using less computation?

**Questions:**

See weakness

---

### Official Review · Reviewer_bcpp · 2025-10-30

**Soundness:** 2
**Presentation:** 3
**Contribution:** 3
**Rating:** 4
**Confidence:** 4

**Summary:**

The paper introduces ACPO, a reinforcement learning framework for aligning vision-language models in complex reasoning tasks. ACPO addresses the limitations of static PPO-style optimization by combining two adaptive mechanisms: 1. a dynamic curriculum that transitions from stable on-policy exploration to efficient off-policy exploitation through progressive sample reuse; 2. an Advantage-Aware Adaptive Clipping strategy that adjusts update bounds per token based on normalized advantage, allowing larger updates for strong signals while constraining noisy ones. Experiments on benchmarks such as MathVerse, LogicVista, and MMMU-Pro show that ACPO achieves improved performance, faster convergence and good stability over baselines like DAPO and PAPO.

**Strengths:**

1. The paper is clearly written and well-structured. The mathematical formulations are easy to follow, and Algorithm 1 concisely summarizes the training pipeline.

2. The algorithmic design is well-motivated and clearly implemented, with thorough ablation studies verifying the contributions of each module.

**Weaknesses:**

1. The title and narrative emphasize multimodal reasoning, but ACPO itself is modality-agnostic. The algorithm operates purely at the policy optimization level without leveraging any vision-specific mechanisms. This creates a mild mismatch between motivation and actual contribution.

2. Both dynamic curriculum learning and adaptive clipping have prior foundations in RL literature. The paper's innovation lies in combining them rather than introducing a new strategies.

3. Since ACPO is not inherently multimodal, additional experiments on pure language tasks would be valuable to verify its generality.

**Questions:**

See above weaknesses.

---

### Official Review · Reviewer_C5H4 · 2025-10-31

**Soundness:** 2
**Presentation:** 2
**Contribution:** 2
**Rating:** 4
**Confidence:** 5

**Summary:**

This paper proposes Adaptive Curriculum Policy Optimization (ACPO), a reinforcement learning framework designed to better align vision-language models (VLMs) in complex reasoning tasks. ACPO introduces two key mechanisms: a dynamic curriculum that transitions from on-policy to off-policy training, and an Advantage-Aware Adaptive Clipping (AAAC) strategy that adjusts update magnitude based on token-level advantage values. The authors evaluate ACPO on several multimodal reasoning benchmarks (e.g., MathVerse, We-Math, LogicVista, MMMU-Pro) using Qwen2.5-VL models of 3B and 7B sizes, demonstrating consistent performance improvements over prior PPO-based baselines.

**Strengths:**

- The proposed adaptive curriculum bridges early-stage stability and late-stage efficiency, which is practically relevant for large-scale policy optimization.
- The AAAC mechanism provides a principled way to handle heterogeneous token-level learning signals, improving both training stability and learning effectiveness.

**Weaknesses:**

- The proposed ACPO framework does not rely on any vision-specific components, yet all experiments are conducted on VLMs. This raises the question of whether ACPO’s claimed benefits generalize to purely LLMs.
- According to Tables 1 and 2, ACPO offers marginal or even negative gains over existing baselines (e.g., PAPO) on 7B-scale models, particularly in vision-dependent multimodal reasoning tasks. Moreover, the comparisons omit several strong and recent baselines [1, 2, 3].
- The AAAC mechanism depends heavily on the choice of δ; inappropriate δ values lead to unstable training. Similarly, the data filtering process (e.g., τ and N_max for B_valid) may require task-specific tuning, limiting general applicability.
- The paper focuses exclusively on quantitative results without showing qualitative examples (e.g., reasoning traces or failure cases) to illustrate how ACPO changes model behavior.

[1]. SoTA with Less: MCTS-Guided Sample Selection for Data-Efficient Visual Reasoning Self-Improvement. NeurIPS 2025.

[2]. VL-Rethinker: Incentivizing Self-Reflection of Vision-Language Models with Reinforcement Learning. NeurIPS 2025.

[3]. NoisyRollout: Reinforcing Visual Reasoning with Data Augmentation. NeurIPS 2025.

**Questions:**

See above

---

### Official Review · Reviewer_2Sk6 · 2025-10-31

**Soundness:** 2
**Presentation:** 3
**Contribution:** 2
**Rating:** 4
**Confidence:** 3

**Summary:**

The paper focuses on the problem of aligning large vision-language models (VLMs) for complex multimodal reasoning tasks using reinforcement learning. The authors point out that standard policy optimization methods have problems with static training schedules and a strict uniform clipping threshold, which can make it harder for models to learn.
To overcome these issues, the paper proposes Adaptive Curriculum Policy Optimization (ACPO), a framework with two key innovations. First, ACPO employs a dynamic curriculum that gradually transitions the training from a near on-policy exploration phase to an off-policy exploitation phase by progressively increasing the sample reuse count during training. This allows the model to start with stable learning on fresh data and later capitalize on valuable past experiences as training progresses. Second, it introduces an Advantage-Aware Adaptive Clipping (AAAC) mechanism, which replaces PPO’s fixed clipping parameter with a dynamic, sample-wise clipping bound modulated by each token’s normalized advantage. In essence, tokens with higher advantage (indicating more beneficial learning signal) are allowed a looser update constraint, enabling larger policy updates, while low-advantage or risky samples get a tighter bound to prevent destructive updates.

**Strengths:**

# Strengths

1.  The proposed ACPO framework introduces a two-pronged adaptive strategy that is novel and intuitive. The dynamic curriculum (adaptive sample reuse) provides a principled way to balance exploration vs. exploitation over the course of training.

2. ACPO delivers strong empirical results. It outperforms state-of-the-art baselines (DAPO, PAPO) on multiple complex reasoning benchmarks, achieving the highest average accuracy in both vision-dependent and general multimodal reasoning categories.

**Weaknesses:**

# Weaknesses

1. Incremental Innovation: While effective, the contributions of ACPO could be viewed as incremental improvements over existing methods rather than entirely new techniques. The idea of curriculum learning in RL (gradually increasing difficulty or sample reuse) is not brand-new.

2. The proposed method can also be used for textual QA tasks. The motivation for only applying on VLM is not clear.

3. The performance gains (based on Tables 1 and 2) are not significant. This makes me question whether the method will consistently underperform the DAPO baseline on text QA tasks.

**Questions:**

# Questions

1. Could you elaborate on the choice of a linear schedule for K(t) (the sample reuse count)? Did you try other scheduling strategies (e.g., exponential increase, or an adaptively determined schedule based on performance)?

2. Did the authors evaluate the proposed model on standard text QA benchmarks—the same settings used in the DAPO paper?

---

### Official Review · Reviewer_daoo · 2025-11-01

**Soundness:** 2
**Presentation:** 2
**Contribution:** 2
**Rating:** 2
**Confidence:** 3

**Summary:**

ACPO is a novel VLM finetuning method by taking inspiration from curriculum learning. It is difficult to finetune a VLM directly on difficult reasoning tasks. To counteract this, ACPO uses an adaptive sampler to create a curriculum that reflects the difficulties of the training task. This allows the policy to use off-policy RL to finetune on these tasks, which gives faster convergence and more training stability on top of empirical gains. When evaluated across 4 difficult reasoning benchmarks, ACPO achieves strong performance when compared against DAPO (both on and off policy) and PAPO.

**Strengths:**

1. I believe there is sufficient coverage in the experiments conducted as it deals with a variety of different difficult reasoning domains.
2. The algorithm section helped me to understand the pipeline well, and I generally like the presentation of the paper.
3. There is good innovation in combining curriculum learning with finetuning for reasoning: I believe that it might be tricky to directly perform finetuning on difficult visual tasks, so it is good to see inclusion of curriculum learning on that end.

**Weaknesses:**

1. There are not enough statistical analyses being performed on the main results and I cannot distinguish whether the results that you have presented are statistically significant. I believe that it will be very helpful for readers if the authors include metrics such as standard error and/or t-tests.
2. The method’s generalizability is being limited due to how many moving components the method employs. In particular, I find the three stages of finetuning (on policy exploration $\rightarrow$ off-policy exploitation) might be very cumbersome. I might be wrong here, but are there any previous works that demonstrate the portability of these implementation details w.r.t. finetuning VLMs?
3. I believe that the writing can be improved. For example, neither $o$ nor $\tau$ were introduced in the main text before their usage in your formulation. Defining each term before using it can strengthen the paper’s presentation.

**Questions:**

1. Instead of using error functions, did you try using other nonlinearities that have the same range as erf, such as tanh or rescaled sigmoid?
2. The abstract mentioned that ACPO is able to achieve accelerated convergence, yet I did not see much of this in figure 2. Can you point to me where did your experiments help validate this claim, or if not, can you provide more ablations to this?
3. What reward models did the authors use?
4. Following up on (3), if there are particular failure cases of the reward model, would there be concerns about overfitting or fitting the VLM to spurious outputs by the reward model?

Minor remarks:
1. In figure 1, $i-th$ and $(i+1)-th$ should be reformatted to $i$-th and $(i+1)$-th to reduce ambiguity.
2. I took a look at the supplementary material and it appears that some of the dataset in visual reasoning are in Chinese. Did the evaluation procedure require outputs from multiple languages? If so, it would be good to include this disclaimer in the main text.

---

### Note · Program_Chairs · 2026-01-17
**Submission Desk Rejected by Program Chairs**

The following references in this submission do not refer to real documents and/or have major errors in bibliographic information:

 Yuhao Zhang, Ge Gao, Zixuan Ma, and et al. Dynamath: Solving math word problems via dynamic evolution of reasoning graphs. arXiv preprint arXiv:2402.11306, 2024b.